# FastSpeech: Fast, Robust and Controllable Text to Speech

**Yi Ren**[*]
Zhejiang University
rayeren@zju.edu.cn

**Yangjun Ruan**[*]
Zhejiang University
ruanyj3107@zju.edu.cn

**Xu Tan**
Microsoft Research
xuta@microsoft.com

**Tao Qin**
Microsoft Research
taoqin@microsoft.com

**Sheng Zhao**
Microsoft STC Asia
Sheng.Zhao@microsoft.com

**Zhou Zhao**[†]
Zhejiang University
zhaozhou@zju.edu.cn

**Tie-Yan Liu**
Microsoft Research
tyliu@microsoft.com

## Abstract

Neural network based end-to-end text to speech (TTS) has significantly improved the quality of synthesized speech. Prominent methods (e.g., Tacotron 2) usually first generate mel-spectrogram from text, and then synthesize speech from the mel-spectrogram using vocoder such as WaveNet. Compared with traditional concatenative and statistical parametric approaches, neural network based end-to-end models suffer from slow inference speed, and the synthesized speech is usually not robust (i.e., some words are skipped or repeated) and lack of controllability (voice speed or prosody control). In this work, we propose a novel feed-forward network based on Transformer to generate mel-spectrogram in parallel for TTS. Specifically, we extract attention alignments from an encoder-decoder based teacher model for phoneme duration prediction, which is used by a length regulator to expand the source phoneme sequence to match the length of the target mel-spectrogram sequence for parallel mel-spectrogram generation. Experiments on the LJSpeech dataset show that our parallel model matches autoregressive models in terms of speech quality, nearly eliminates the problem of word skipping and repeating in particularly hard cases, and can adjust voice speed smoothly. Most importantly, compared with autoregressive Transformer TTS, our model speeds up mel-spectrogram generation by 270x and the end-to-end speech synthesis by 38x. Therefore, we call our model FastSpeech. [3]

## 1 Introduction

Text to speech (TTS) has attracted a lot of attention in recent years due to the advance in deep learning. Deep neural network based systems have become more and more popular for TTS, such as Tacotron [27], Tacotron 2 [22], Deep Voice 3 [19], and the fully end-to-end ClariNet [18]. Those models usually first generate mel-spectrogram autoregressively from text input and then synthesize speech from the mel-spectrogram using vocoder such as Griffin-Lim [6], WaveNet [24], Parallel

---

[*]Equal contribution.

[†]Corresponding author

[3]Synthesized speech samples can be found in `https://speechresearch.github.io/fastspeech/`.

WaveNet [16], or WaveGlow [20][4]. Neural network based TTS has outperformed conventional concatenative and statistical parametric approaches [9, 28] in terms of speech quality.

In current neural network based TTS systems, mel-spectrogram is generated autoregressively. Due to the long sequence of the mel-spectrogram and the autoregressive nature, those systems face several challenges:

- Slow inference speed for mel-spectrogram generation. Although CNN and Transformer based TTS [14, 19] can speed up the training over RNN-based models [22], all models generate a mel-spectrogram conditioned on the previously generated mel-spectrograms and suffer from slow inference speed, given the mel-spectrogram sequence is usually with a length of hundreds or thousands.

- Synthesized speech is usually not robust. Due to error propagation [3] and the wrong attention alignments between text and speech in the autoregressive generation, the generated mel-spectrogram is usually deficient with the problem of words skipping and repeating [19].

- Synthesized speech is lack of controllability. Previous autoregressive models generate mel-spectrograms one by one automatically, without explicitly leveraging the alignments between text and speech. As a consequence, it is usually hard to directly control the voice speed and prosody in the autoregressive generation.

Considering the monotonous alignment between text and speech, to speed up mel-spectrogram generation, in this work, we propose a novel model, FastSpeech, which takes a text (phoneme) sequence as input and generates mel-spectrograms non-autoregressively. It adopts a feed-forward network based on the self-attention in Transformer [25] and 1D convolution [5, 11, 19]. Since a mel-spectrogram sequence is much longer than its corresponding phoneme sequence, in order to solve the problem of length mismatch between the two sequences, FastSpeech adopts a length regulator that up-samples the phoneme sequence according to the phoneme duration (i.e., the number of mel-spectrograms that each phoneme corresponds to) to match the length of the mel-spectrogram sequence. The regulator is built on a phoneme duration predictor, which predicts the duration of each phoneme.

Our proposed FastSpeech can address the above-mentioned three challenges as follows:

- Through parallel mel-spectrogram generation, FastSpeech greatly speeds up the synthesis process.

- Phoneme duration predictor ensures hard alignments between a phoneme and its mel-spectrograms, which is very different from soft and automatic attention alignments in the autoregressive models. Thus, FastSpeech avoids the issues of error propagation and wrong attention alignments, consequently reducing the ratio of the skipped words and repeated words.

- The length regulator can easily adjust voice speed by lengthening or shortening the phoneme duration to determine the length of the generated mel-spectrograms, and can also control part of the prosody by adding breaks between adjacent phonemes.

We conduct experiments on the LJSpeech dataset to test FastSpeech. The results show that in terms of speech quality, FastSpeech nearly matches the autoregressive Transformer model. Furthermore, FastSpeech achieves 270x speedup on mel-spectrogram generation and 38x speedup on final speech synthesis compared with the autoregressive Transformer TTS model, almost eliminates the problem of word skipping and repeating, and can adjust voice speed smoothly. We attach some audio files generated by our method in the supplementary materials.

## 2   Background

In this section, we briefly overview the background of this work, including text to speech, sequence to sequence learning, and non-autoregressive sequence generation.

**Text to Speech** TTS [1, 18, 21, 22, 27], which aims to synthesize natural and intelligible speech given text, has long been a hot research topic in the field of artificial intelligence. The research on TTS has shifted from early concatenative synthesis [9], statistical parametric synthesis [13, 28] to neural network based parametric synthesis [1] and end-to-end models [14, 18, 22, 27], and the quality of the synthesized speech by end-to-end models is close to human parity. Neural network based end-to-end TTS models usually first convert the text to acoustic features (e.g., mel-spectrograms) and then transform mel-spectrograms into audio samples. However, most neural TTS systems generate mel-spectrograms autoregressively, which suffers from slow inference speed, and synthesized speech usually lacks of robustness (word skipping and repeating) and controllability (voice speed or prosody control). In this work, we propose FastSpeech to generate mel-spectrograms non-autoregressively, which sufficiently handles the above problems.

**Sequence to Sequence Learning** Sequence to sequence learning [2, 4, 25] is usually built on the encoder-decoder framework: The encoder takes the source sequence as input and generates a set of representations. After that, the decoder estimates the conditional probability of each target element given the source representations and its preceding elements. The attention mechanism [2] is further introduced between the encoder and decoder in order to find which source representations to focus on when predicting the current element, and is an important component for sequence to sequence learning.

In this work, instead of using the conventional encoder-attention-decoder framework for sequence to sequence learning, we propose a feed-forward network to generate a sequence in parallel.

**Non-Autoregressive Sequence Generation** Unlike autoregressive sequence generation, non-autoregressive models generate sequence in parallel, without explicitly depending on the previous elements, which can greatly speed up the inference process. Non-autoregressive generation has been studied in some sequence generation tasks such as neural machine translation [7, 8, 26] and audio synthesis [16, 18, 20]. Our FastSpeech differs from the above works in two aspects: 1) Previous works adopt non-autoregressive generation in neural machine translation or audio synthesis mainly for inference speedup, while FastSpeech focuses on both inference speedup and improving the robustness and controllability of the synthesized speech in TTS. 2) For TTS, although Parallel WaveNet [16], ClariNet [18] and WaveGlow [20] generate audio in parallel, they are conditioned on mel-spectrograms, which are still generated autoregressively. Therefore, they do not address the challenges considered in this work. There is a concurrent work [17] that also generates mel-spectrogram in parallel. However, it still adopts the encoder-decoder framework with attention mechanism, which 1) requires 2~3x model parameters compared with the teacher model and thus achieves slower inference speedup than FastSpeech; 2) cannot totally solve the problems of word skipping and repeating while FastSpeech nearly eliminates these issues.

## 3 FastSpeech

In this section, we introduce the architecture design of FastSpeech. To generate a target mel-spectrogram sequence in parallel, we design a novel feed-forward structure, instead of using the encoder-attention-decoder based architecture as adopted by most sequence to sequence based autoregressive [14, 22, 25] and non-autoregressive [7, 8, 26] generation. The overall model architecture of FastSpeech is shown in Figure 1. We describe the components in detail in the following subsections.

### 3.1 Feed-Forward Transformer

The architecture for FastSpeech is a feed-forward structure based on self-attention in Transformer [25] and 1D convolution [5, 19]. We call this structure as Feed-Forward Transformer (FFT), as shown in Figure 1a. Feed-Forward Transformer stacks multiple FFT blocks for phoneme to mel-spectrogram transformation, with $N$ blocks on the phoneme side, and $N$ blocks on the mel-spectrogram side, with a length regulator (which will be described in the next subsection) in between to bridge the length gap between the phoneme and mel-spectrogram sequence. Each FFT block consists of a self-attention and 1D convolutional network, as shown in Figure 1b. The self-attention network consists of a multi-head attention to extract the cross-position information. Different from the 2-layer dense network in Transformer, we use a 2-layer 1D convolutional network with ReLU activation. The motivation is that the adjacent hidden states are more closely related in the character/phoneme and mel-spectrogram

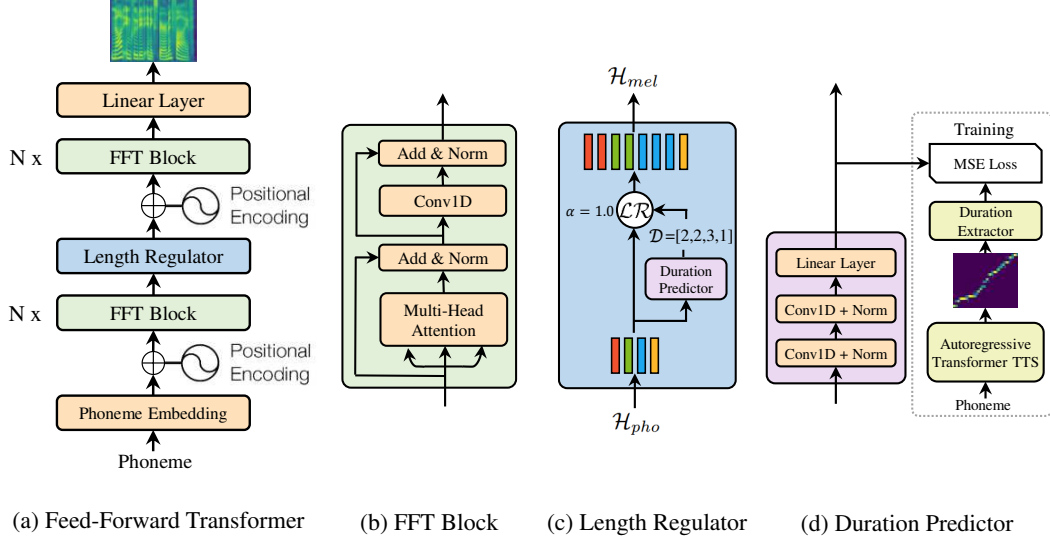

(a) Feed-Forward Transformer     (b) FFT Block     (c) Length Regulator     (d) Duration Predictor

Figure 1: The overall architecture for FastSpeech. (a). The feed-forward Transformer. (b). The feed-forward Transformer block. (c). The length regulator. (d). The duration predictor. MSE loss denotes the loss between predicted and extracted duration, which only exists in the training process.

sequence in speech tasks. We evaluate the effectiveness of the 1D convolutional network in the experimental section. Following Transformer [25], residual connections, layer normalization, and dropout are added after the self-attention network and 1D convolutional network respectively.

## 3.2 Length Regulator

The length regulator (Figure 1c) is used to solve the problem of length mismatch between the phoneme and spectrogram sequence in the Feed-Forward Transformer, as well as to control the voice speed and part of prosody. The length of a phoneme sequence is usually smaller than that of its mel-spectrogram sequence, and each phoneme corresponds to several mel-spectrograms. We refer to the length of the mel-spectrograms that corresponds to a phoneme as the phoneme duration (we will describe how to predict phoneme duration in the next subsection). Based on the phoneme duration $d$, the length regulator expands the hidden states of the phoneme sequence $d$ times, and then the total length of the hidden states equals the length of the mel-spectrograms. Denote the hidden states of the phoneme sequence as $\mathcal{H}_{pho} = [h_1, h_2, ..., h_n]$, where $n$ is the length of the sequence. Denote the phoneme duration sequence as $\mathcal{D} = [d_1, d_2, ..., d_n]$, where $\Sigma_{i=1}^n d_i = m$ and $m$ is the length of the mel-spectrogram sequence. We denote the length regulator $\mathcal{LR}$ as

$$\mathcal{H}_{mel} = \mathcal{LR}(\mathcal{H}_{pho}, \mathcal{D}, \alpha), \tag{1}$$

where $\alpha$ is a hyperparameter to determine the length of the expanded sequence $\mathcal{H}_{mel}$, thereby controlling the voice speed. For example, given $\mathcal{H}_{pho} = [h_1, h_2, h_3, h_4]$ and the corresponding phoneme duration sequence $\mathcal{D} = [2, 2, 3, 1]$, the expanded sequence $\mathcal{H}_{mel}$ based on Equation 1 becomes $[h_1, h_1, h_2, h_2, h_3, h_3, h_3, h_4]$ if $\alpha = 1$ (normal speed). When $\alpha = 1.3$ (slow speed) and $0.5$ (fast speed), the duration sequences become $\mathcal{D}_{\alpha=1.3} = [2.6, 2.6, 3.9, 1.3] \approx [3, 3, 4, 1]$ and $\mathcal{D}_{\alpha=0.5} = [1, 1, 1.5, 0.5] \approx [1, 1, 2, 1]$, and the expanded sequences become $[h_1, h_1, h_1, h_2, h_2, h_2, h_3, h_3, h_3, h_3, h_4]$ and $[h_1, h_2, h_3, h_3, h_4]$ respectively. We can also control the break between words by adjusting the duration of the space characters in the sentence, so as to adjust part of prosody of the synthesized speech.

## 3.3 Duration Predictor

Phoneme duration prediction is important for the length regulator. As shown in Figure 1d, the duration predictor consists of a 2-layer 1D convolutional network with ReLU activation, each followed by the layer normalization and the dropout layer, and an extra linear layer to output a scalar, which is exactly the predicted phoneme duration. Note that this module is stacked on top of the FFT blocks on the phoneme side and is jointly trained with the FastSpeech model to predict the length of

mel-spectrograms for each phoneme with the mean square error (MSE) loss. We predict the length in the logarithmic domain, which makes them more Gaussian and easier to train. Note that the trained duration predictor is only used in the TTS inference phase, because we can directly use the phoneme duration extracted from an autoregressive teacher model in training (see following discussions).

In order to train the duration predictor, we extract the ground-truth phoneme duration from an autoregressive teacher TTS model, as shown in Figure 1d. We describe the detailed steps as follows:

- We first train an autoregressive encoder-attention-decoder based Transformer TTS model following [14].

- For each training sequence pair, we extract the decoder-to-encoder attention alignments from the trained teacher model. There are multiple attention alignments due to the multi-head self-attention [25], and not all attention heads demonstrate the diagonal property (the phoneme and mel-spectrogram sequence are monotonously aligned). We propose a focus rate $F$ to measure how an attention head is close to diagonal: $F = \frac{1}{S} \sum_{s=1}^{S} \max_{1 \leq t \leq T} a_{s,t}$, where $S$ and $T$ are the lengths of the ground-truth spectrograms and phonemes, $a_{s,t}$ donates the element in the $s$-th row and $t$-th column of the attention matrix. We compute the focus rate for each head and choose the head with the largest $F$ as the attention alignments.

- Finally, we extract the phoneme duration sequence $\mathcal{D} = [d_1, d_2, ..., d_n]$ according to the duration extractor $d_i = \sum_{s=1}^{S} [\arg\max_t a_{s,t} = i]$. That is, the duration of a phoneme is the number of mel-spectrograms attended to it according to the attention head selected in the above step.

# 4 Experimental Setup

## 4.1 Datasets

We conduct experiments on LJSpeech dataset [10], which contains 13,100 English audio clips and the corresponding text transcripts, with the total audio length of approximate 24 hours. We randomly split the dataset into 3 sets: 12500 samples for training, 300 samples for validation and 300 samples for testing. In order to alleviate the mispronunciation problem, we convert the text sequence into the phoneme sequence with our internal grapheme-to-phoneme conversion tool [23], following [1, 22, 27]. For the speech data, we convert the raw waveform into mel-spectrograms following [22]. Our frame size and hop size are set to 1024 and 256, respectively.

In order to evaluate the robustness of our proposed FastSpeech, we also choose 50 sentences which are particularly hard for TTS system, following the practice in [19].

## 4.2 Model Configuration

**FastSpeech model**  Our FastSpeech model consists of 6 FFT blocks on both the phoneme side and the mel-spectrogram side. The size of the phoneme vocabulary is 51, including punctuations. The dimension of phoneme embeddings, the hidden size of the self-attention and 1D convolution in the FFT block are all set to 384. The number of attention heads is set to 2. The kernel sizes of the 1D convolution in the 2-layer convolutional network are both set to 3, with input/output size of 384/1536 for the first layer and 1536/384 in the second layer. The output linear layer converts the 384-dimensional hidden into 80-dimensional mel-spectrogram. In our duration predictor, the kernel sizes of the 1D convolution are set to 3, with input/output sizes of 384/384 for both layers.

**Autoregressive Transformer TTS model**  The autoregressive Transformer TTS model serves two purposes in our work: 1) to extract the phoneme duration as the target to train the duration predictor; 2) to generate mel-spectrogram in the sequence-level knowledge distillation (which will be introduced in the next subsection). We refer to [14] for the configurations of this model, which consists of a 6-layer encoder, a 6-layer decoder, except that we use 1D convolution network instead of position-wise FFN. The number of parameters of this teacher model is similar to that of our FastSpeech model.

### 4.3 Training and Inference

We first train the autoregressive Transformer TTS model on 4 NVIDIA V100 GPUs, with batchsize of 16 sentences on each GPU. We use the Adam optimizer with $\beta_1 = 0.9$, $\beta_2 = 0.98$, $\varepsilon = 10^{-9}$ and follow the same learning rate schedule in [25]. It takes 80k steps for training until convergence. We feed the text and speech pairs in the training set to the model again to obtain the encoder-decoder attention alignments, which are used to train the duration predictor. In addition, we also leverage sequence-level knowledge distillation [12] that has achieved good performance in non-autoregressive machine translation [7, 8, 26] to transfer the knowledge from the teacher model to the student model. For each source text sequence, we generate the mel-spectrograms with the autoregressive Transformer TTS model and take the source text and the generated mel-spectrograms as the paired data for FastSpeech model training.

We train the FastSpeech model together with the duration predictor. The optimizer and other hyper-parameters for FastSpeech are the same as the autoregressive Transformer TTS model. The FastSpeech model training takes about 80k steps on 4 NVIDIA V100 GPUs. In the inference process, the output mel-spectrograms of our FastSpeech model are transformed into audio samples using the pretrained WaveGlow [20][5].

## 5 Results

In this section, we evaluate the performance of FastSpeech in terms of audio quality, inference speedup, robustness, and controllability.

**Audio Quality** We conduct the MOS (mean opinion score) evaluation on the test set to measure the audio quality. We keep the text content consistent among different models so as to exclude other interference factors, only examining the audio quality. Each audio is listened by at least 20 testers, who are all native English speakers. We compare the MOS of the generated audio samples by our FastSpeech model with other systems, which include 1) *GT*, the ground truth audio; 2) *GT (Mel + WaveGlow)*, where we first convert the ground truth audio into mel-spectrograms, and then convert the mel-spectrograms back to audio using WaveGlow; 3) *Tacotron 2 [22] (Mel + WaveGlow)*; 4) *Transformer TTS [14] (Mel + WaveGlow)*. 5) *Merlin [28] (WORLD)*, a popular parametric TTS system with WORLD [15] as the vocoder. The results are shown in Table 1. It can be seen that our FastSpeech can nearly match the quality of the Transformer TTS model and Tacotron 2 [6].

| Method | MOS |
|---|---|
| *GT* | $4.41 \pm 0.08$ |
| *GT (Mel + WaveGlow)* | $4.00 \pm 0.09$ |
| *Tacotron 2 [22] (Mel + WaveGlow)* | $3.86 \pm 0.09$ |
| *Merlin [28] (WORLD)* | $2.40 \pm 0.13$ |
| *Transformer TTS [14] (Mel + WaveGlow)* | $3.88 \pm 0.09$ |
| *FastSpeech (Mel + WaveGlow)* | $3.84 \pm 0.08$ |

Table 1: The MOS with 95% confidence intervals.

**Inference Speedup** We evaluate the inference latency of FastSpeech compared with the autoregressive Transformer TTS model, which has similar number of model parameters with FastSpeech. We first show the inference speedup for mel-spectrogram generation in Table 2. It can be seen that FastSpeech speeds up the mel-spectrogram generation by 269.40x, compared with the Transformer TTS model. We then show the end-to-end speedup when using WaveGlow as the vocoder. It can be seen that FastSpeech can still achieve 38.30x speedup for audio generation.

| Method | Latency (s) | Speedup |
|---|---|---|
| *Transformer TTS [14] (Mel)* | 6.735 ± 3.969 | / |
| *FastSpeech (Mel)* | 0.025 ± 0.005 | 269.40× |
| *Transformer TTS [14] (Mel + WaveGlow)* | 6.895 ± 3.969 | / |
| *FastSpeech (Mel + WaveGlow)* | 0.180 ± 0.078 | 38.30× |

Table 2: The comparison of inference latency with 95% confidence intervals. The evaluation is conducted on a server with 12 Intel Xeon CPU, 256GB memory, 1 NVIDIA V100 GPU and batch size of 1. The average length of the generated mel-spectrograms for the two systems are both about 560.

We also visualize the relationship between the inference latency and the length of the predicted mel-spectrogram sequence in the test set. Figure 2 shows that the inference latency barely increases with the length of the predicted mel-spectrogram for FastSpeech, while increases largely in Transformer TTS. This indicates that the inference speed of our method is not sensitive to the length of generated audio due to parallel generation.

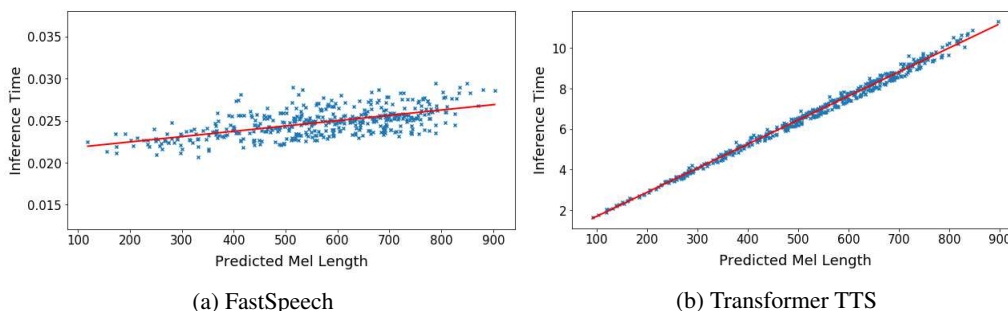

(a) FastSpeech  (b) Transformer TTS

Figure 2: Inference time (second) vs. mel-spectrogram length for FastSpeech and Transformer TTS.

**Robustness** The encoder-decoder attention mechanism in the autoregressive model may cause wrong attention alignments between phoneme and mel-spectrogram, resulting in instability with word repeating and word skipping. To evaluate the robustness of FastSpeech, we select 50 sentences which are particularly hard for TTS system[7]. Word error counts are listed in Table 3. It can be seen that Transformer TTS is not robust to these hard cases and gets 34% error rate, while FastSpeech can effectively eliminate word repeating and skipping to improve intelligibility.

| Method | Repeats | Skips | Error Sentences | Error Rate |
|---|---|---|---|---|
| *Tacotron 2* | 4 | 11 | 12 | 24% |
| *Transformer TTS* | 7 | 15 | 17 | 34% |
| *FastSpeech* | 0 | 0 | 0 | 0% |

Table 3: The comparison of robustness between FastSpeech and other systems on the 50 particularly hard sentences. Each kind of word error is counted at most once per sentence.

**Length Control** As mentioned in Section 3.2, FastSpeech can control the voice speed as well as part of the prosody by adjusting the phoneme duration, which cannot be supported by other end-to-end TTS systems. We show the mel-spectrograms before and after the length control, and also put the audio samples in the supplementary material for reference.

*Voice Speed* The generated mel-spectrograms with different voice speeds by lengthening or shortening the phoneme duration are shown in Figure 3. We also attach several audio samples in the

supplementary material for reference. As demonstrated by the samples, FastSpeech can adjust the voice speed from 0.5x to 1.5x smoothly, with stable and almost unchanged pitch.

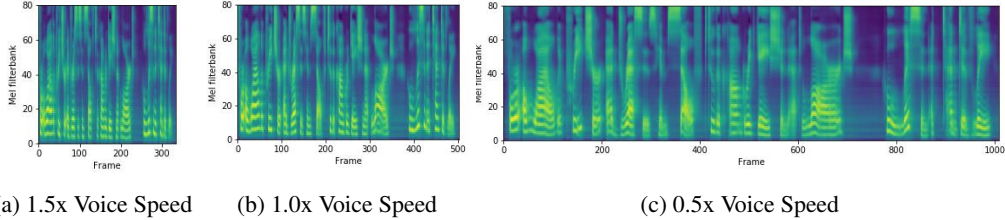

(a) 1.5x Voice Speed  (b) 1.0x Voice Speed  (c) 0.5x Voice Speed

Figure 3: The mel-spectrograms of the voice with 1.5x, 1.0x and 0.5x speed respectively. The input text is "*For a while the preacher addresses himself to the congregation at large, who listen attentively*".

*Breaks Between Words* FastSpeech can add breaks between adjacent words by lengthening the duration of the space characters in the sentence, which can improve the prosody of voice. We show an example in Figure 4, where we add breaks in two positions of the sentence to improve the prosody.

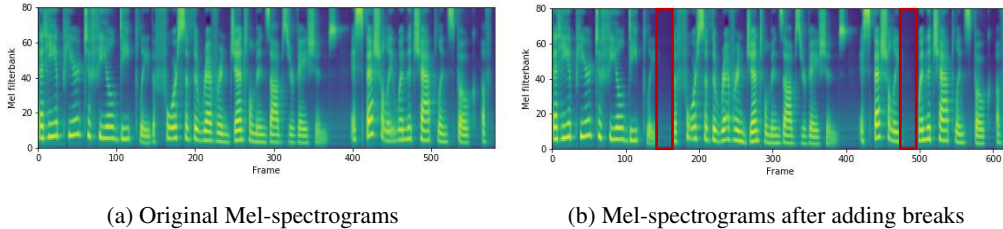

(a) Original Mel-spectrograms  (b) Mel-spectrograms after adding breaks

Figure 4: The mel-spectrograms before and after adding breaks between words. The corresponding text is "*that he appeared to feel **deeply** the force of the reverend gentleman's observations, **especially** when the chaplain spoke of*". We add breaks after the words "*deeply*" and "*especially*" to improve the prosody. The red boxes in Figure 4b correspond to the added breaks.

**Ablation Study**   We conduct ablation studies to verify the effectiveness of several components in FastSpeech, including 1D Convolution and sequence-level knowledge distillation. We conduct CMOS evaluation for these ablation studies.

| System | CMOS |
|---|---|
| *FastSpeech* | 0 |
| *FastSpeech without 1D convolution in FFT block* | -0.113 |
| *FastSpeech without sequence-level knowledge distillation* | -0.325 |

Table 4: CMOS comparison in the ablation studies.

*1D Convolution in FFT Block* We propose to replace the original fully connected layer (adopted in Transformer [25]) with 1D convolution in FFT block, as described in Section 3.1. Here we conduct experiments to compare the performance of 1D convolution to the fully connected layer with similar number of parameters. As shown in Table 4, replacing 1D convolution with fully connected layer results in -0.113 CMOS, which demonstrates the effectiveness of 1D convolution.

*Sequence-Level Knowledge Distillation* As described in Section 4.3, we leverage sequence-level knowledge distillation for FastSpeech. We conduct CMOS evaluation to compare the performance of FastSpeech with and without sequence-level knowledge distillation, as shown in Table 4. We find that removing sequence-level knowledge distillation results in -0.325 CMOS, which demonstrates the effectiveness of sequence-level knowledge distillation.

# 6 Conclusions

In this work, we have proposed FastSpeech: a fast, robust and controllable neural TTS system. FastSpeech has a novel feed-forward network to generate mel-spectrogram in parallel, which consists of several key components including feed-forward Transformer blocks, a length regulator and a duration predictor. Experiments on LJSpeech dataset demonstrate that our proposed FastSpeech can nearly match the autoregressive Transformer TTS model in terms of speech quality, speed up the mel-spectrogram generation by 270x and the end-to-end speech synthesis by 38x, almost eliminate the problem of word skipping and repeating, and can adjust voice speed (0.5x-1.5x) smoothly.

For future work, we will continue to improve the quality of the synthesized speech, and apply FastSpeech to multi-speaker and low-resource settings. We will also train FastSpeech jointly with a parallel neural vocoder to make it fully end-to-end and parallel.

## Acknowledgments

This work was supported by the National Natural Science Foundation of China under Grant No.61602405, No.61836002. This work was also supported by the China Knowledge Centre of Engineering Sciences and Technology.

## Footnotes

[4]Although ClariNet [18] is fully end-to-end, it still first generates mel-spectrogram autoregressively and then synthesizes speech in one model.

[5]https://github.com/NVIDIA/waveglow

[6]According to our further comprehensive experiments on our internal datasets, the voice quality of FastSpeech can always match that of the teacher model on multiple languages and multiple voices, if we use more unlabeled text for knowledge distillation.

[7]These cases include single letters, spellings, repeated numbers, and long sentences. We list the cases in the supplementary materials.

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
