[Supplementary Material · appendices.pdf]

# Appendices

## A   Model Hyperparameters

Table 5: Hyperparameters of Transformer TTS and FastSpeech. Encoder and Decoder are for Transformer TTS, Phoneme-Side and Mel-Side FFT are for FastSpeech.

| Hyperparameter | Transformer TTS | FastSpeech |
|---|---|---|
| Phoneme Embedding Dimension | 512 | 384 |
| Encoder/Phoneme-Side FFT Layers | 6 | 6 |
| Encoder/Phoneme-Side FFT Hidden | 512 | 384 |
| Encoder/Phoneme-Side FFT Conv1D Kernel | 1 | 3 |
| Encoder/Phoneme-Side FFT Conv1D Filter Size | 1024 | 1536 |
| Encoder/Phoneme-Side FFT Attention Heads | 8 | 2 |
| Decoder/Mel-Side FFT Layers | 6 | 6 |
| Decoder/Mel-Side FFT Hidden | 512 | 384 |
| Decoder/Mel-Side FFT Conv1D Kernel | 1 | 3 |
| Decoder/Mel-Side FFT Conv1D Filter Size | 1024 | 1536 |
| Decoder/Mel-Side FFT Attention Headers | 8 | 2 |
| Duration Predictor Conv1D Kernel | / | 3 |
| Duration Predictor Conv1D Filter Size | / | 256 |
| Dropout | 0.1 | 0.1 |
| Batch Size | 64 (16 * 4GPUs) | 64 (16 * 4GPUs) |
| Total Number of Parameters | 33.8M | 30.1M |

## B   50 Particularly Hard Sentences

The 50 particularly hard sentences mentioned in Section 5 are listed below:

01. a
02. b
03. c
04. H
05. I
06. J
07. K
08. L
09. 22222222 hello 22222222
10. S D S D Pass zero - zero Fail - zero to zero - zero - zero Cancelled - fifty nine to three - two - sixty four Total - fifty nine to three - two -
11. S D S D Pass - zero - zero - zero - zero Fail - zero - zero - zero - zero Cancelled - four hundred and sixteen - seventy six -
12. zero - one - one - two Cancelled - zero - zero - zero - zero Total - two hundred and eighty six - nineteen - seven -
13. forty one to five three hundred and eleven Fail - one - one to zero two Cancelled - zero - zero to zero zero Total -
14. zero zero one , MS03 - zero twenty five , MS03 - zero thirty two , MS03 - zero thirty nine ,
15. 1b204928 zero zero zero zero zero zero zero zero zero zero zero zero zero zero one seven ole32
16. zero zero zero zero zero zero zero zero two seven nine eight F three forty zero zero zero zero zero six four two eight zero one eight
17. c five eight zero three three nine a zero bf eight FALSE zero zero zero bba3add2 - c229 - 4cdb -

18. Calendaring agent failed with error code 0x80070005 while saving appointment .

19. Exit process - break ld - Load module - output ud - Unload module - ignore ser - System error - ignore ibp - Initial breakpoint -

20. Common DB connectors include the DB - nine , DB - fifteen , DB - nineteen , DB - twenty five , DB - thirty seven , and DB - fifty connectors .

21. To deliver interfaces that are significantly better suited to create and process RFC eight twenty one , RFC eight twenty two , RFC nine seventy seven , and MIME content .

22. int1 , int2 , int3 , int4 , int5 , int6 , int7 , int8 , int9 ,

23. seven _ ctl00 ctl04 ctl01 ctl00 ctl00

24. Http0XX , Http1XX , Http2XX , Http3XX ,

25. config file must contain A , B , C , D , E , F , and G .

26. mondo - debug mondo - ship motif - debug motif - ship sts - debug sts - ship Comparing local files to checkpoint files ...

27. Rusbvts . dll Dsaccessbvts . dll Exchmembvt . dll Draino . dll Im trying to deploy a new topology , and I keep getting this error .

28. You can call me directly at four two five seven zero three seven three four four or my cell four two five four four four seven four seven four or send me a meeting request with all the appropriate information .

29. Failed zero point zero zero percent < one zero zero one zero zero zero zero zero Internal . Exchange . ContentFilter . BVT ContentFilter . BVT_log . xml Error ! Filename not specified .

30. C colon backslash o one two f c p a r t y backslash d e v one two backslash oasys backslash legacy backslash web backslash HELP

31. src backslash mapi backslash t n e f d e c dot c dot o l d backslash backslash m o z a r t f one backslash e x five

32. copy backslash backslash j o h n f a n four backslash scratch backslash M i c r o s o f t dot S h a r e P o i n t dot

33. Take a look at h t t p colon slash slash w w w dot granite dot a b dot c a slash access slash email dot

34. backslash bin backslash premium backslash forms backslash r e g i o n a l o p t i o n s dot a s p x dot c s Raj , DJ ,

35. Anuraag backslash backslash r a d u r five backslash d e b u g dot one eight zero nine underscore P R two h dot s t s contains

36. p l a t f o r m right bracket backslash left bracket f l a v o r right bracket backslash s e t u p dot e x e

37. backslash x eight six backslash Ship backslash zero backslash A d d r e s s B o o k dot C o n t a c t s A d d r e s

38. Mine is here backslash backslash g a b e h a l l hyphen m o t h r a backslash S v r underscore O f f i c e s v r

39. h t t p colon slash slash teams slash sites slash T A G slash default dot aspx As always , any feedback , comments ,

40. two thousand and five h t t p colon slash slash news dot com dot com slash i slash n e slash f d slash two zero zero three slash f d

41. backslash i n t e r n a l dot e x c h a n g e dot m a n a g e m e n t dot s y s t e m m a n a g e

42. I think Rich's post highlights that we could have been more strategic about how the sum total of XBOX three hundred and sixtys were distributed .

43. 64X64 , 8K , one hundred and eighty four ASSEMBLY , DIGITAL VIDEO DISK DRIVE , INTERNAL , 8X ,

44. So we are back to Extended MAPI and C++ because . Extended MAPI does not have a dual interface VB or VB .Net can read .

45. Thanks , Borge Trongmo Hi gurus , Could you help us E2K ASP guys with the following issue ?

46. Thanks J RGR Are you using the LDDM driver for this system or the in the build XDDM driver ?

47. Btw , you might remember me from our discussion about OWA automation and OWA readiness day a year ago .

48. empidtool . exe creates HKEY_CURRENT_USER Software Microsoft Office Common QMPersNum in the registry , queries AD , and the populate the registry with MS employment ID if available else an error code is logged .

49. Thursday, via a joint press release and Microsoft AI Blog, we will announce Microsoft's continued partnership with Shell leveraging cloud, AI, and collaboration technology to drive industry innovation and transformation.

50. Actress Fan Bingbing attends the screening of 'Ash Is Purest White (Jiang Hu Er Nv)' during the 71st annual Cannes Film Festival