[Reviews · NeurIPS 2019]

Reviewer 1



Originally: Although phoneme duration prediction is widely adopted in conventional TTS systems, jointly training it in a neural TTS model is new. This paper is one of the first works on non-autoregressive text-to-spectrogram modeling. Quality: This paper seems sound overall, expected for a few issues in the comments below. Some of these issues must be addressed before acceptance. Clarity: A well written paper. A good reading to me, except for a few comments below. Significance: The advantages over its autoregressive counterparts are significant, especially for industrial use. It’s likely to be followed by the research community as well as the industry. Comments: 1. It’s interesting to see the using of 2-head attention for the Transformer block, instead of more popular setting ups such as 8 in the baseline Transformer TTS model. Does it bring benefits? 2. What’s the reason for using the mel-spectrogram generated by the autoregressive model for distillation training, instead of using the groundtruth mel-spectrogram? Intuitively, the groundtruth gives more accurate information. 3. Sec. 4.3 says that the FastSpeech model is partially initialized from the autoregressive Transformer TTS model (phoneme embeddings and FFT blocks) as they share the same architecture. However, the hyperparams given in Appendix A as well as in Sec. 4.2 shows these two models are of different dimensions for these components. 4. The pre-net and post-net of the baseline autoregressive Transformer TTS, as well as the decoder’s final linear layer of FastSpeech seem missing from the hyperparams comparison in Appendix A. 5. Experiment results on inference speedup -- what’s the batch size used for this evaluation? 6. The latency numbers in Table 2 and Figure 2 seem inconsistent. The numbers in Table 2 seem unrealistically fast. 7. Robustness experiment -- Since you have included Tacotron 2 in Table 1, it would be nice to also include Tacotron 2 in Table 3. Tacotron 2 is another widely discussed attention-based model which is considered also suffering from robustness issues due to attention failure. It will be interesting to include such results for comparison. 8. Needs a reference for CMOS evaluation. ============== Update: Thanks for authors' response. I updated my score accordingly.

Reviewer 2



Authors propose a non auto-regressive parallel text 2 mel-spectrogram model that allows a significant speed up in text to speech generation. The underlying model is based on feed forward transformer model extended with two auxiliary tasks for predicting length and duration of the underlying phonemes (i.e the input is phoneme, not word-based). To appropriately train the whole system the approach still requires auto regressive teacher model to properly work out phoneme durations. The model does not seem sensitive to spurious generative errors like repetitions or omissions. For waveform generation another non-autoregressive waveglow vocoder is used (not a contribution). Overall the study seems fair and reproducible, proposes several solutions for existing shortcomings in e2e TTS systems, offers large speedup while preserves accuracy of auto-regressive models. It's a good paper. Would it make a difference if you assumed access to pronunciation dictionary? G2p may introduce some errors along the way (though good it works with it regardless). You could probably enforce monotonicity in the attention aligner, rather than score them based on which one behaves as you hope. I would definitely cite [1] as you borrow a number of blocks and ideas from that paper (like 1d convolutions in tts context, etc.) [1] FFTNET: A REAL-TIME SPEAKER-DEPENDENT NEURAL VOCODER, 2018 Minor: Several unnecessary repetitions. ==== Update: Thanks for answering my concerns. Wrt g2p thing - you should make it explicit in the paper your system requires a pronunciation dictionary.

Reviewer 3



Comments: 1. The audio quality and inference speedup are impressive. 2. In session 3.2 Length Regulator, the hidden states of the phoneme sequence are simply repeated, which is very much like what Gu et al. did in "Non-autoregressive neural machine translation". However, the advantage of attention-based sequence-to-sequence speech synthesis model is the soft alignments between phonemes and spectrograms. Empirically, the soft attention gives better prosody and more natural speech. Won't the hard alignments(rounding and repetition) hurt the performance of the proposed model? 3. In session 3.3 Duration Predictor, the proposed focus rate F has nothing to do with "measuring how an attention head is close to diagonal". Focus rate sort of measures the overall confidence of attention alignments, but doesn't constrain the attention alignments close to diagonal. Also, it's hard to understand the behavior of each head in multihead attention, and in many cases, the attention doesn't have any clear visual meanings at all. Then why diagonal alignments are good and what if there is NO diagonal alignments in multihead attention? 4. The title is improper. In TTS, "controllable" always means the prosody or pitch diversity under expressive settings. In session 5, the voice-speed control and breaks-between-words control are trivial to TTS models. It shouldn't be called "controllable".

[Author Response · NeurIPS 2019]

Thanks all the reviewers for the comments and suggestions!

**To Reviewer #1**

- **About 2-head attention** Compared with the attention alignments in machine translation, the attention in TTS is monotonous and much simpler than machine translation. Therefore, we choose fewer attention heads. Our preliminary experiments also show that more heads do not bring much difference to Transformer TTS. We will report the numbers in the new version.

- **Generated v.s. Groundtruth mel-spectrogram** Knowledge distillation is widely used in non-autoregressive machine translation [1] and speech synthesis [2] for transfer knowledge from the autoregressive teacher model to the non-autoregressive student model. The intuitive explanation is that the teacher model can generate data with smoother distribution and less noise, which is easy to be fitted by the student model [1][2]. The ablation study in Table 4 demonstrates the effectiveness of knowledge distillation.

- **Unmatched hyperparams** Our FastSpeech model differs from the original Transformer TTS model in that we use Conv1D instead of the original dense network after multi-head attention. So in our experiment, FastSpeech model is initialized from the teacher model with the same configuration, but not the original Transformer TTS model as shown in Appendix A. We will explicitly point out this in the new version of the paper.

- **Missing hyperparams in Appendix** Thanks for your reminder. We will add the missing hyperparams of pre-net and post-net in the new version of paper. For Transformer TTS, the hidden dimension and the number of layers of CNN are 512, 5 for post-net, and are 512, 3 for pre-net. For FastSpeech, the hidden dimension of the decoder's final linear layer is 80.

- **Batch size for inference** The batch size is 1 for inference evaluation, in order to simulate the scenario of online production. Many previous works (e.g., [1][2][3]) use this batch size to evaluate the inference latency.

- **About the inconsistent latency numbers** There is a typo in Table 2. The unit of latency should be "second". We will fix it in the new version of the paper.

- **Robustness test for Tacotron 2** We evaluate the robustness of Tacotron 2 on the 50 hard sentences. The repeating, skipping and error sentences are 4, 11 and 12 respectively, and the error rate is 24%. We will add the results in the new version of the paper.

- **The reference for CMOS evaluation** We will add the reference for CMOS in the new version of the paper.

[1] Gu, Jiatao, et al. "Non-autoregressive neural machine translation." ICLR 2018.

[2] Oord, Aaron van den, et al. "Parallel wavenet: Fast high-fidelity speech synthesis." ICML 2018.

[3] Prenger, Ryan, et al. "Waveglow: A flow-based generative network for speech synthesis." ICASSP 2019.

**To Reviewer #2**

- **About pronunciation dictionary** Our grapheme-to-phoneme tool works like this: it first looks up the pronunciation dictionary, and if the dictionary does not contain the word, it predicts the phoneme using a grapheme-to-phoneme model.

- **Enforce monotonicity** Thanks for your suggestion. We observe from the experiments that there are roughly two kinds of attention in Transformer TTS: diagonal and non-diagonal. If we enforce all the attentions to be diagonal, the performance of Transformer TTS will drop, because the non-diagonal attentions are also very helpful to the model. So we select the diagonal attentions rather than enforce all attentions to be monotonic.

- **Add the citation** We will add this citation in the new version of the paper.

**To Reviewer #3**

- **About the hard alignments** We've tried to add Gaussian smoothing to soften the inputs of the FFT block in the mel side. However, it doesn't improve the performance. We think the Conv1D and self-attention can mix the information from neighbors and play a role in softening the hard copied hidden states.

- **About diagonal alignments** We visualize the attention of each head in Transformer TTS and find that nearly half of the attentions are always diagonal for each data pair. We also find that if an attention has high focus rate, it is always diagonal. We find that some errors are closely related to the poor attention: jumping attention causes skipping and non-monotonic attention causes repeating. We think that diagonal attentions can ensure correct alignments in transformer TTS. We could not find any cases that have no diagonal attention at all.

- **About the title** Thanks for your advice! We will change the title accordingly.

[Meta-Review · NeurIPS 2019]

The paper proposes a novel non-autoregressive parallelisation approach for mel-spectrogram intermediate representation TTS. The reviewers concur that the paper incorporates two novel explicit components to tts systems - length and duration modules and that the results on Speedup at inference and high-quality audio generations are relevant.